# US and EU Free Trade Agreements and implementation of policies to control tobacco, alcohol, and unhealthy food and drinks: A quasi-experimental analysis

**Pepita Barlow**[1]*, **Luke N. Allen**[2]

**1** Department of Health Policy, London School of Economics and Political Science, London, United Kingdom, **2** Department of Clinical Research, London School of Hygiene and Tropical Medicine, London, United Kingdom

\* p.barlow@lse.ac.uk

## Abstract

### Background

Identifying and tackling the factors that undermine regulation of unhealthy commodities is an essential component of effective noncommunicable disease (NCD) prevention. Unhealthy commodity producers may use rules in US and EU Free Trade Agreements (FTAs) to challenge policies targeting their products. We aimed to test whether there was a statistical relationship between US and EU FTA participation and reduced implementation of WHO-recommended policies.

### Methods and findings

We performed a statistical analysis assessing the probability of at least partially implementing 10 tobacco, alcohol, and unhealthy food and drink policies in 127 countries in 2014, 2016, and 2019. We assessed differences in implementation of these policies in countries with and without US/EU FTAs. We used matching to conduct 48 covariate-adjusted quasi-experimental comparisons across 27 matched US/EU FTA members (87 country-years) and performed additional analyses and robustness checks to assess alternative explanations for our results. Out of our 48 tests, 19% (9/48) identified a statistically significant decrease in the predicted probability of at least partially implementing the unhealthy commodity policy in question, while 2% (1/48) showed an increase. However, there was marked heterogeneity across policies. At the level of individual policies, US FTA participation was associated with a 37% reduction (95%CI: −0.51 to −0.22) in the probability of fully implementing graphic tobacco warning policies, and a 53% reduction (95%CI: −0.63 to −0.43) in the probability of at least partially implementing smoke-free place policies. EU FTA participation was associated with a 28% reduction (95%CI: −0.45 to −0.10) in the probability of fully implementing graphic tobacco warning policies, and a 25% reduction (95%CI: −0.47 to −0.03) in the probability of fully implementing restrictions on child marketing of unhealthy food and drinks. There was a positive association with implementing fat limits and bans, but this was not robust. Associations with other outcomes were not significant. The main

**Data Availability Statement:** All data files are available from the Harvard Dataverse database: Barlow, Pepita, 2022, "Replication Data for: US and EU Free Trade Agreements and implementation of

policies to control tobacco, alcohol, and unhealthy food and drink: a quasi-experimental analysis", https://doi.org/10.7910/DVN/THBIDB, Harvard Dataverse.

**Funding:** The authors received no specific funding for this work.

**Competing interests:** The authors have declared that no competing interests exist.

**Abbreviations:** AME, average marginal effect; BIT, Bilateral Investment Treaty; CP-TPP, Comprehensive and Progressive Agreement for Trans-Pacific Partnership; FTA, Free Trade Agreement; NCD, noncommunicable disease; SDG, Sustainable Development Goal.

limitations included residual confounding, limited ability to discern precise mechanisms of influence, and potentially limited generalisability to other FTAs.

## Conclusions

US and EU FTA participation may reduce the probability of implementing WHO-recommended tobacco and child food marketing policies by between a quarter and a half—depending on the FTA and outcome in question. Governments negotiating or participating in US/EU FTAs may need to establish robust health protections and mitigation strategies to achieve their NCD mortality reduction targets.

---

## Author summary

### Why was this study done?

- Identifying and attending to the factors that inhibit the proper regulation of unhealthy commodities is a pressing priority for governments seeking to accelerate progress towards reducing noncommunicable diseases (NCDs).

- US and EU Free Trade Agreements (FTAs) may play a significant role in stalling policy progress by incentivising and empowering unhealthy commodity producers to challenge policies targeting their products in FTA partner countries.

- However, these agreements also acknowledge governments' right to regulate and protect public health, and previous studies were unable to establish whether countries with US/EU FTAs are typically less successful at implementing unhealthy commodity policies.

### What did the researchers do and find?

- We conducted a global statistical analysis assessing the relationship between US and EU FTA participation and implementation of WHO-recommended policies targeting unhealthy commodities.

- Our large-scale quantitative approach allows for the incorporation of data from many more countries and time periods than previous approaches while addressing key alternative explanations in our main models and >30 additional analyses and robustness checks.

- We identified a substantial reduction in the predicted probability of implementing select WHO-recommended policies in countries participating in US FTAs and EU FTAs, with the probability of implementing tobacco and child food marketing restrictions reducing by between a quarter and a half depending on the FTA and regulation in question; other associations were not significant.

### What do these findings mean?

- Our findings indicate that participating in US and EU FTAs is associated with reduced implementation of select unhealthy commodity policies that are crucial to achieving

global targets to prevent and reduce NCD-related mortality, morbidity, and associated treatment costs.

- For countries currently negotiating US/EU FTAs, there is now a potential opportunity to ensure these agreements do not empower unhealthy commodity producers to challenge unhealthy commodity policies and instead empower governments to accelerate NCD policy progress.

- For countries already participating in US/EU FTAs, governments will need to ensure their policies are not unduly influenced by vested interests that are often concealed in technical discussions about trade rules.

## Introduction

Noncommunicable diseases (NCDs) are responsible for more than 70% of global deaths [1,2], and Sustainable Development Goal (SDG) target 3.4 calls for all Member States to reduce premature NCD mortality by a third between 2015 and 2030 [3]. By 2019, over 40% of WHO-recommended NCD policies had been implemented worldwide, including physical activity mass media campaigns, national NCD action plans, national NCD targets, and clinical guidelines for addressing NCDs (Fig 1). However, relatively few countries have adopted WHO-backed policies targeting the marketing, sale, and consumption of tobacco, alcohol, and unhealthy

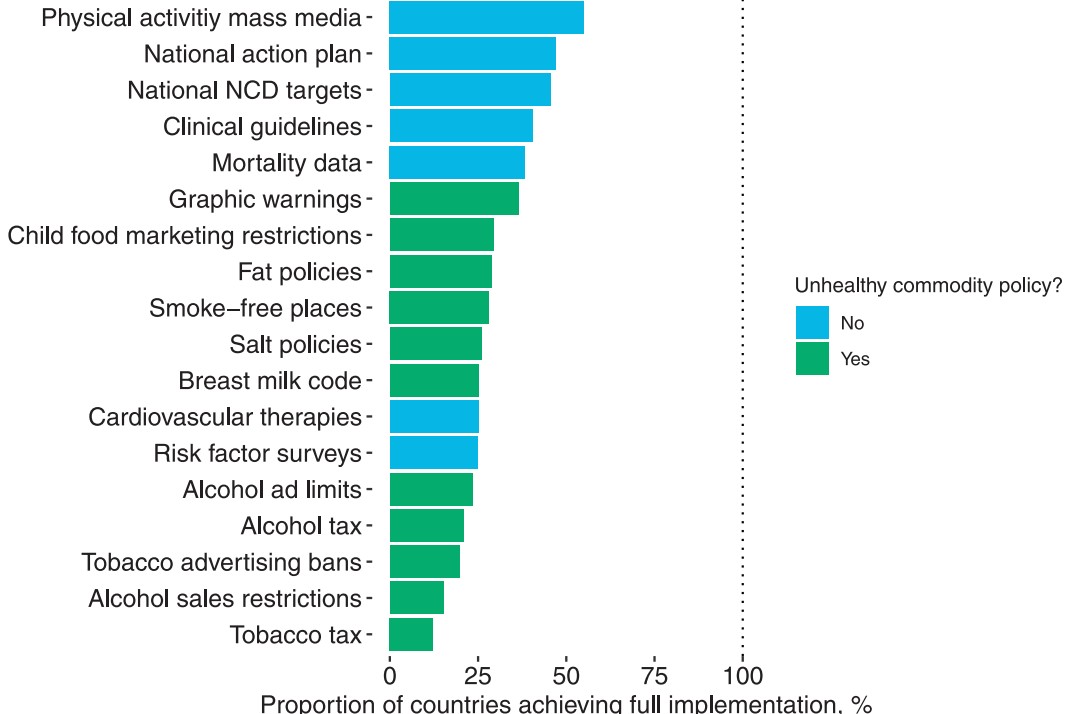

**Fig 1. Global implementation of NCD policies in 2019.** Notes: Data from Allen and colleagues based on WHO NCD Country Capacity Surveys [4,29,30].

food and drinks; "unhealthy commodities" [4,5]. These policies are essential for achieving targets to reduce NCD mortality, morbidity, and associated clinical treatment costs [6]. One recent analysis of 21 WHO-recommended policies to address NCDs found that as much as two-thirds of the total mortality impact of the policy package could be achieved through policies targeting unhealthy commodities alone [6].

Policy-makers, academics, and civil society have long noted that producers of unhealthy commodities play a major role in stalling and undermining policy progress by forcefully opposing policies targeting their products [7]. However, industry opposition does not occur in vacuum, as it can be sustained by the treaties, institutions, activities, and norms outside the health sector that create opportunities for business input and influence on policy [8,9]. It is essential to assess how such opportunities for industry opposition are established, as these same avenues can be targeted in order to accelerate policy progress [10].

US and EU Free Trade Agreements (FTAs) create opportunities for the large, multinational tobacco, food, and alcohol companies headquartered in these jurisdictions to oppose unhealthy commodity policies, as summarised in Fig 2 [11–13]. FTAs are important instruments of foreign economic integration that are designed to reduce cross-border trade and investment costs [14]. By June 2022, 355 FTAs were in force globally, representing a 17-fold increase since 1991 [15]. While globalisation appears to be decelerating, several US and EU FTAs are now under negotiation including, for example, the US–Kenya and US–UK deals, and EU agreements with New Zealand, the Philippines, Indonesia, China, and Australia [15,16]. The UK is also negotiating accession to the Comprehensive and Progressive

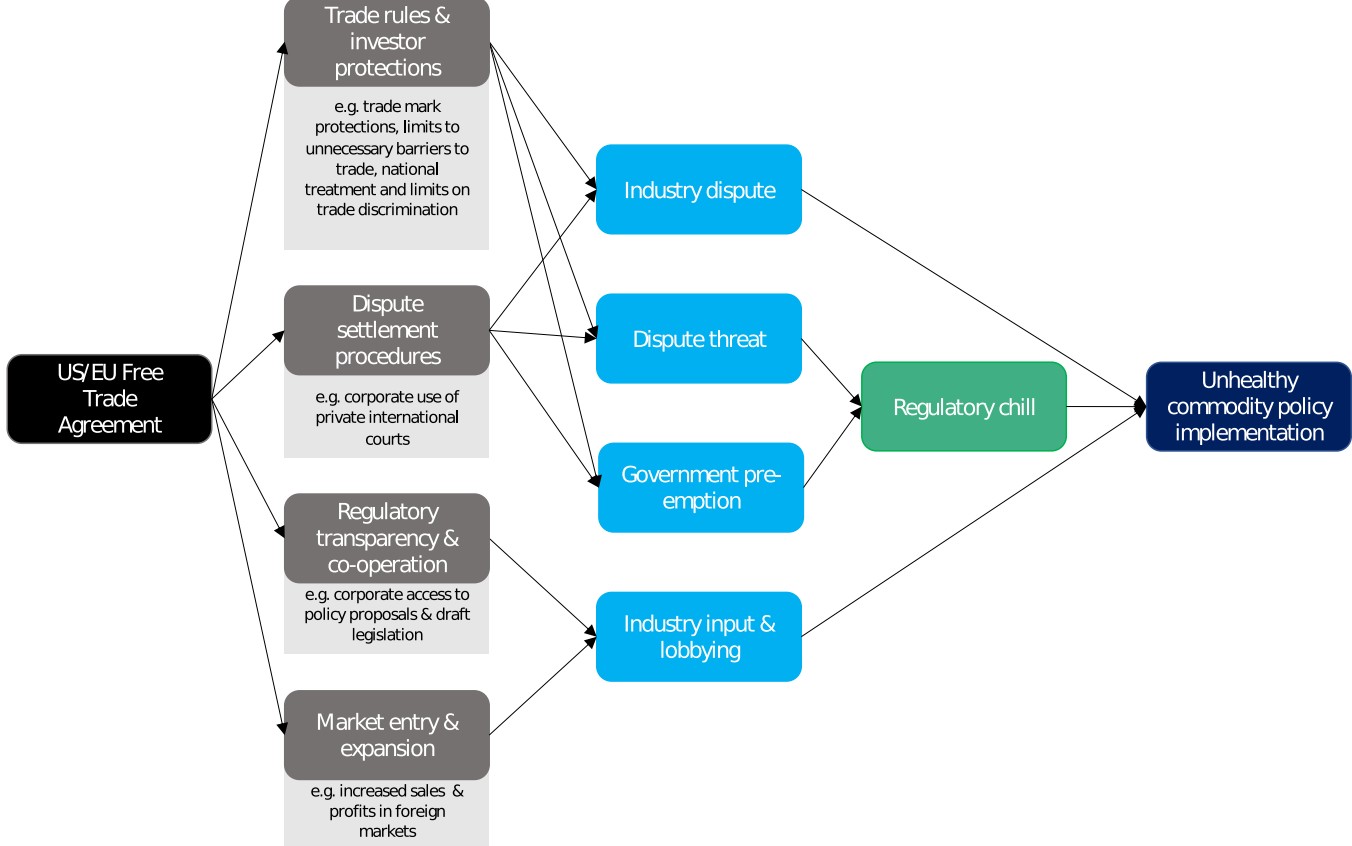

**Fig 2. Logic model summarising how US and EU FTAs influence unhealthy commodity policy implementation.**

Agreement for Trans-Pacific Partnership, CP-TPP, which was heavily influenced by the US before it withdrew from the agreement. Domestic policies, such as those targeting unhealthy commodities, can create trade and business costs, and so FTAs increasingly incorporate rules covering domestic policies, known as "deep" or "behind the border" clauses [14] There is variation in FTA design, but those negotiated by the US and EU include typically include stringent domestic policy-making clauses and grant extensive investor protections and intellectual property rights [17,18]. Many large unhealthy commodity producers are also headquartered in the US and EU.

A substantial body of scholarship documents the various ways that firms can and have used FTA clauses to influence policies targeting their products, summarised in Box 1 and Fig 2 [11–13,19–21].

A recent systematic review of FTAs and health did not identify any quantitative studies that systematically examined the relationship between participation in US and EU FTAs and unhealthy commodity policy implementation [22]. Previous small-N studies instead identified instances when food, tobacco, and alcohol companies threatened or initiated trade disputes about policies affecting their products [13,23–25]. There are also ex ante impact assessments identifying the mechanisms through which specific FTAs may impede policy implementation [12,26,27]. In-depth case studies further connected select challenges to policy outcomes, including instances of policy modification and delayed implementation [13,23–25]. However,

---

### Box 1. US/EU Free Trade Agreements and unhealthy commodity policy implementation.

US/EU FTAs contain written recognition that governments have a legitimate right to protect public health. They also establish trade rules, investor protections, and dispute settlement procedures that unhealthy commodity producers can use to pressure governments to modify or abandon a policy initiating or threatening a trade **dispute**. A range of relevant clauses can be cited, including those protecting investors against an "expropriation" of their investments. Businesses might claim this occurs where a policy affects a businesses' profitability, for example, due to the impact of marketing, sales, and consumption restrictions on product sales. Additional clauses include, for example, those setting extensive intellectual property protections, which might be used to challenge marketing restrictions or labelling policies that affect the use of trademarks or brands.

**Dispute threats** are an especially common form of industry pressure and can also be particularly effective in prompting governments to abandon a regulation or adopt industry-preferred alternatives if governments seek to avoid the costs of a dispute—so-called "**regulatory chill.**" Importantly, **dispute threats** can be influential even where they are at odds with legal analyses. Furthermore, even the perceived risk of such a threat or dispute from industry can prompt governments to **preemptively change** their policies.

US/EU FTAs also enable and incentivise direct influence and **lobbying** as they can include regulatory cooperation clauses that enable private sector stakeholders to scrutinise policy proposals and directly influence a diversity of policies as they are being developed. In addition, US/EU FTAs foster market entry and expansion by multinational unhealthy commodity producers. This may increase their lobbying activities due to their incentive to protect sales and profits in FTA–partner jurisdictions.

case studies of policy change may constitute exceptions to a broader tendency for governments with US/EU FTAs to successfully implement unhealthy commodity policies, especially because these agreements also acknowledge governments' right to regulate and protect public health [19]. In some contexts, FTA participation may also promote policy implementation where companies seek to harmonise policies across trade borders towards a higher regulatory standard [28]. It remains unclear whether the potential impacts identified in ex ante FTA appraisals have materialised, ex post, for countries participating in US and EU FTAs.

Here we conduct a statistical analysis assessing the relationship between US/EU FTA participation and the implementation of WHO-recommended policies on tobacco, alcohol, and unhealthy food and drinks.

## Methods

### Data

We assessed the relationship between US/EU FTA participation and the achievement of (i) partial or full implementation; and (ii) full implementation of policies targeting tobacco, alcohol, and unhealthy food and drinks in 127 countries with available covariate data in 2014, 2016, and 2019. We analysed 10 categories of WHO-recommended policies including taxes and restrictions on marketing, sales, and consumption, as described in Box 2. Implementation of these policies is assessed by WHO using regular NCD Country Capacity Surveys [29]. Cross-sectional implementation survey responses were published by WHO and systematically coded by Allen and colleagues [4,30]. We combined this information with US/EU FTA participation data from the Design of Trade Agreements Database and covariate data from multiple sources (see Table A in S1 Supporting Information for full list and rationale) [17]. All economic data were adjusted for inflation and purchasing power.

### Statistical models

Following a published protocol [31], we used matching—a quasi-experimental approach—to help account for nonrandom assignment into FTAs. Matching preprocesses the data by identifying a subset of comparable countries from the overall pool of observations [32–34]. Further analysis is then performed using only comparable matched sets. Unlike regression adjustment, matching can increase the internal validity of causal estimates by increasing the comparability of the "control group" of countries that did not participate in US or EU FTA, while increasing the transparency of any residual differences in covariate values and the precise counterfactual contrasts used to estimate causal effects [32,33]. We examined the performance of multiple available matching algorithms and measures of comparability between countries, and present results from the model with the largest number of successfully matched countries and the best performance on covariate balance tests, on average, in models assessing both US FTAs and EU FTAs (see Appendix A in S1 Supporting Information).

Our final models used full matching on the Mahalanobis distance, a composite measure of the differences in the characteristics of countries with and without US/EU FTAs [32]. The full matching algorithm places each country with (or without) US/EU FTAs into subsets with at least 1 country without (or with) a US/EU FTA with the smallest Mahalanobis distance(s) from the country with the US/EU FTA [35]. The algorithm further ensures that the final sum of the Mahalanobis distances across all matched sets is minimised. To improve comparability, we restricted comparisons to observations in the same year and WTO membership status and limited differences in GDP per capita between units with and without US/EU FTAs to USD10,000. This identified matches for up to $n = 15$ countries with US FTAs (45 country-years) and $n = 12$ successful matches (36 country-years) for EU FTAs.

## Box 2. Policies targeting the marketing, composition, and consumption of unhealthy commodities.

Tobacco

- Tobacco taxes

- Smoke-free place policies

- Graphic warnings on cigarette packages

- Tobacco advertising bans

Alcohol

- Alcohol sales or advertising restrictions

- Alcohol taxes

Unhealthy food and nonalcoholic beverages

- Legislation implementing the International Code of Marketing of Breastmilk Substitutes.

- Policies to reduce salt/ sodium consumption

- Policies to limit saturated fatty acids and eliminate *trans*-fats

- Policies targeting the marketing of foods and nonalcoholic beverages to children

Notes: All categories listed above correspond to those originally captured in WHO country surveys and categorised therein, with the exception of alcohol sales and advertising restrictions. We grouped these into a single category as very few countries had implemented these policies and we sought to ensure there was variation in implementation across FTA partners. For each policy above, we create 2 dichotomous indicators capturing at least partial (i.e., partial or full) implementation of the regulation, and full implementation of the regulation.

We first estimated logistic regression models using unmatched data with controls for covariates, which may influence US or EU FTA participation and unhealthy commodity policies (see Appendix A in S1 Supporting Information). We then generated the matched subsets using these same covariates to measure differences in unit characteristics (summarised in the Mahalanobis distance). We then performed a series of covariate balance tests to assess the performance of our chosen matching algorithm in reducing differences in the characteristics of countries with and without US or EU FTAs, before and after matching (see Appendix A and Table C in S1 Supporting Information). Next, we reestimated our regression models with controls for any covariates that remain imbalanced in the matched subsets, as indicated by an absolute standardised difference in means across countries with/without US/EU FTAs larger than 0.1.

Our baseline regression models are as follows:

Equation 1. US agreements.

$$\text{Implementation}_{it} = B_0 + B_1 \text{USFTA}_{it-1} + B_2 X_{it-1} + B_3 \text{Wave}_t + e_{it}$$

Equation 2. EU agreements.

$$Implementation_{it} = B_0 + B_1 EUFTA_{it-1} + B_2 X_{it-1} + B_3 Wave_t + e_{it}$$

where Implementation$_{it}$ is one of the binary indicators of partial/full implementation (10 indicators) or full implementation (10 indicators) of a particular policy in country i in year t, and $B_0$ is the intercept. We created 2 dichotomous indicators of participating in either a US or EU FTA: USFTA$_{it-1}$ in Eq 1 captures whether country i participated in a US FTA (1) or not (0) in year t-1, and EUFTA$_{it-1}$ in Eq 2 captures whether country i participated in an EU FTA (1) or not (0) in year t-1. $X_{it-1}$ is a vector of controls in year t-1 with coefficients in the vector $B_2$. In both models, we control for democratisation, GDP per capita, the share of the population of secondary education age that is enrolled in secondary education, implementation of non-trade business policies, WTO participation, geographic region (converted into a series of region dummies), and international political integration (or "political globalisation"). We further control for participation in FTAs with countries other than US/EU where the world's 10 largest producers of tobacco, alcohol, and unhealthy food and drinks are headquartered (see Tables A and B in Supporting Information for full list and measurement of these covariates). Wave$_t$ in Eqs 1 and 2 is a control for the wave of data collection and accounts for unobserved factors that influence implementation, vary time periods, and are common across all countries. $e_{it}$ in Eqs 1 and 2 is the error term. We estimate block-bootstrapped standard errors, which approximate cluster robust standard errors [36–38]. Our bootstrap procedure samples matched strata from the matched sample, where each strata ID contains at least 1 country with a US or EU FTA and at least 1 country without a US or EU FTA in a given year.

Finally, we use the estimated models to calculate average marginal effects (AMEs): differences in the predicted probability of implementation according to US/EU FTA participation status [39]. All models were estimated using R version 4.1.3. This study is reported as per the Strengthening the Reporting of Observational Studies in Epidemiology (STROBE) guideline (S1 STROBE Checklist).

## Results

Table 1 presents descriptive statistics of countries according to their US and EU FTA status before we identified matched subsets. Countries with and without US or EU FTAs significantly differ from one another with respect to several characteristics, including domestic market liberalisation, GDP per capita, WTO membership status, and geographic region. This underscores the importance of performing quasi-experimental contrasts to account for differences in covariate characteristics.

Fig 3 plots the absolute standardised difference in the mean characteristics of countries with and without US/EU FTAs, before and after matching. Fig 3 shows that matching yields large reduction in the differences in the characteristics of countries with and without US/EU FTAs.

Table 2 and Fig 4 show the adjusted results from our matching estimators. Approximately 19% (9/48) of tests showed a statistically significant decrease in the predicted probability of at least partial unhealthy commodity policy implementation, while 2% (1/48) showed an increase. However, other associations were not significant. Thus, on average across all policy outcomes, US FTA participation associated with a 5 percentage point (%) (95% CI: −0.17 to 0.07) decline in the predicted probability of at least partially implementing the unhealthy commodity policy in question, and this association was not statistically significant. Similarly, EU FTA participation was associated, on average, with a 5% (95% CI: −0.11 to 0.01) decline in the

**Table 1. Prematching characteristics of countries with and without US FTAs and EU FTAs.**

| Variable | US FTAs | | | | EU FTAs | | | |
|---|---|---|---|---|---|---|---|---|
| | Without[a] | With | Difference | p-Value[b] | Without | With | Difference | p-Value[b] |
| GDP per capita ($) | 22,705.79 (21,053.97) | 26,201.55 (17,956.99) | −3,496.00 | 0.23 | 24,387.44 (21,672.89) | 15,660.89 (8,527.76) | 8,727.00 | <0.001 |
| Polyarchy score (0–1) | 0.61 (0.25) | 0.62 (0.24) | −0.01 | 0.79 | 0.60 (0.26) | 0.70 (0.13) | −0.11 | <0.001 |
| Domestic market liberalisation index (0–100) | 67.42 (14.56) | 71.16 (11.02) | −3.70 | 0.041 | 68.32 (14.60) | 65.78 (10.49) | 2.50 | 0.17 |
| Proportion with secondary education (%) | 87.06 (30.16) | 94.12 (21.68) | −7.10 | 0.053 | 88.36 (30.23) | 86.56 (20.55) | 1.80 | 0.62 |
| KOF Political globalisation index (0–100) | 73.54 (17.76) | 75.37 (12.63) | −1.80 | 0.39 | 74.36 (17.50) | 70.23 (13.64) | 4.10 | 0.081 |
| WTO membership (0 or 1) | 0.59 (0.49) | 1.00 (0.00) | −0.41 | <0.001 | 0.93 (0.25) | 1.00 (0.00) | −0.07 | <0.001 |
| FTA with another country or countries with large food company HQ(s) (0 or 1) | 0.55 (0.50) | 0.71 (0.46) | −0.17 | 0.02 | 0.60 (0.49) | 0.74 (0.44) | −0.14 | 0.053 |
| FTA with another country or countries with large tobacco company HQ(s) (0 or 1) | 0.55 (0.50) | 0.84 (0.37) | −0.28 | <0.001 | 0.56 (0.50) | 0.63 (0.49) | −0.07 | 0.38 |
| FTA with another country or countries with large alcohol company HQ(s) (0 or 1) | 0.59 (0.49) | 1.00 (0.00) | −0.41 | <0.001 | 0.56 (0.50) | 1.00 (0.00) | −0.44 | <0.001 |
| Region | | | | | | | | |
| East Asia and Pacific | 0.08 (0.27) | 0.12 (0.33) | −0.05 | 0.36 | 0.10 (0.29) | 0.00 (0.00) | 0.10 | <0.001 |
| Europe and Central Asia | 0.45 (0.50) | 0.00 (0.00) | 0.45 | <0.001 | 0.42 (0.49) | 0.14 (0.35) | 0.28 | <0.001 |
| Latin America and Caribbean | 0.08 (0.28) | 0.57 (0.50) | −0.49 | <0.001 | 0.06 (0.24) | 0.79 (0.41) | −0.73 | <0.001 |
| Middle East and North Africa | 0.05 (0.22) | 0.24 (0.43) | −0.19 | 0.003 | 0.09 (0.29) | 0.00 (0.00) | 0.09 | <0.001 |
| North America | 0.01 (0.10) | 0.06 (0.24) | −0.05 | 0.16 | 0.02 (0.13) | 0.02 (0.15) | −0.01 | 0.82 |
| South Asia | 0.07 (0.26) | 0.00 (0.00) | 0.07 | <0.001 | 0.07 (0.26) | 0.00 (0.00) | 0.07 | <0.001 |
| Sub-Saharan Africa | 0.25 (0.44) | 0.00 (0.00) | 0.25 | <0.001 | 0.24 (0.43) | 0.05 (0.21) | 0.19 | <0.001 |
| Sample size (country-years) | 276 | 48 | | | 282 | 43 | | |

FTA, Free Trade Agreement.

[a]Mean or proportion (%).

[b]p-Value for Welch two-sample t test of differences.

For consistency with our statistical models, all variables are lagged by 1 year apart from secondary education, which is lagged by 2 years to ensure sufficient data availability.

predicted probability of at least partially implementing the unhealthy commodity policy in question, and this association was not statistically significant.

There were, however, substantial differences in the association between both US FTA and EU FTA participation and the probability of implementing each policy according to the specific policy and FTA in question. US FTA participation was associated with a 37% lower predicted probability of fully implementing graphic warning policies (95% CI: −0.51 to −0.22), a 24% lower (95% CI: −0.39 to −0.08) probability of fully implementing smoke-free place policies, and a 53% lower (95% CI: −0.63 to −0.43) probability of at least partially implementing the same.

EU FTA participation was associated with a 28% (95%CI: −0.45 to −0.10) lower probability of achieving full implementation of graphic warning policies, and a 25% (95% CI: −0.47 to −0.03) lower probability of fully implementing child marketing restrictions. Comparable associations were observed for partial implementation of graphic warning and child food marketing policies. EU FTA participation was also associated with a 16% (95% CI: 0.01 to 0.03) higher probability of partially implementing fat limits and bans, although this was not robust in additional analyses presented below.

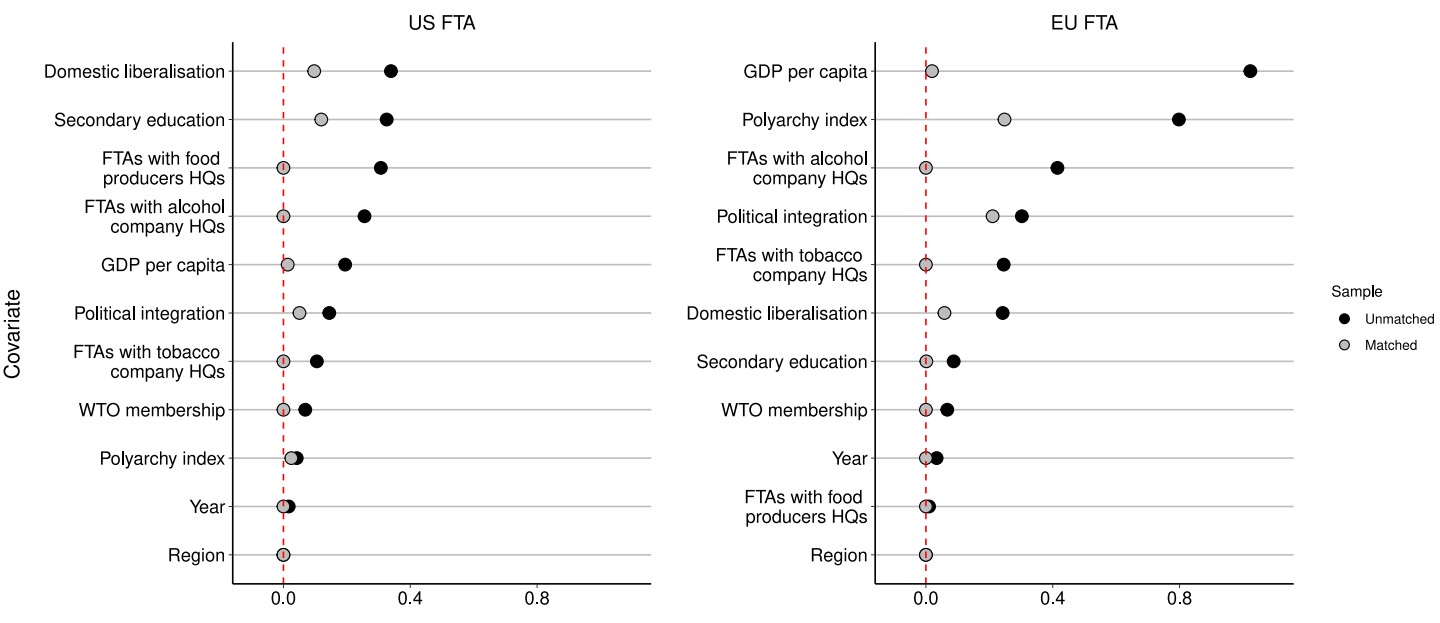

**Fig 3. Covariate balance in matched and unmatched data.**

Prematching regression models are presented in Tables D and E in Supporting Information.

## Additional analyses

In line with our protocol, we conducted 35 additional analyses and robustness checks to assess whether our results may be explained by alternative processes. We first conducted placebo analyses in which we reestimated our models examining implementation of 2 NCD policies that we would not expect to be affected by FTAs: whether or not a country has implemented risk factor surveys or time-bound national targets to address NCDs. Table F in S1 Supporting

**Table 2. AME (95% CI) of US and EU FTA participation on predicted probability of unhealthy commodity policy implementation.**

|  | US FTAs | | EU FTAs | |
|---|---|---|---|---|
|  | **Partial or full implementation** | **Full implementation** | **Partial or full implementation** | **Full implementation** |
| Tobacco taxes | 0.05 (−0.08 to 0.18) | 0.05 (−0.20 to 0.30) | −0.06 (−0.17 to 0.05) | −0.05 (−0.25 to 0.14) |
| Smoke-free places | **−0.53 (−0.63 to −0.43)** | **−0.24 (−0.39 to −0.08)** | −0.05 (−0.16 to 0.07) | 0.14 (−0.11 to 0.39) |
| Graphic warnings | **−0.09 (−0.50 to 0.32)** | **−0.37 (−0.51 to −0.22)** | **−0.25 (−0.36 to −0.13)** | **−0.28 (−0.45 to −0.10)** |
| Tobacco ad bans | 0.01 (−0.16 to 0.18) | 0.05 (−0.19 to 0.29) | −0.07 (−0.31 to 0.16) | −0.04 (−0.27 to 0.18) |
| Alcohol ad and sales restrictions | −0.03 (−0.29 to 0.23) | 0.03 (−0.02 to 0.07) | −0.22 (−0.48 to 0.05) | −0.30 (−0.67 to 0.06) |
| Alcohol taxes | −0.01 (−0.19 to 0.17) | −0.05 (−0.23 to 0.14) | 0.01 (−0.24 to 0.25) | 0.20 (−0.01 to 0.40) |
| Salt reduction | −0.01 (−0.23 to 0.21) | −0.04 (−0.27 to 0.19) | 0.11 (−0.08 to 0.31) | 0.14 (−0.03 to 0.30) |
| Fat limits and bans | −0.04 (−0.24 to 0.16) | −0.07 (−0.26 to 0.12) | **0.16 (0.01 to 0.30)** | 0.11 (−0.06 to 0.28) |
| Child marketing restrictions | 0.12 (−0.06 to 0.30) | 0.12 (−0.06 to 0.30) | **−0.25 (−0.47 to −0.03)** | **−0.25 (−0.47 to −0.03)** |
| Breast milk code | 0.18 (−0.03 to 0.40) | 0.19 (−0.06 to 0.45) | 0.14 (−0.07 to 0.34) | 0.09 (−0.13 to 0.30) |

Boldface indicates statistically significant results. Figures show the difference in the predicted probability of achieving partial/full or full implementation of a given policy among countries with and without either US FTAs (Columns 2–3) or EU FTAs (Columns 4–5). 95% CIs are shown in parentheses.

AME, average marginal effect; CI, confidence interval; FTA, Free Trade Agreement.

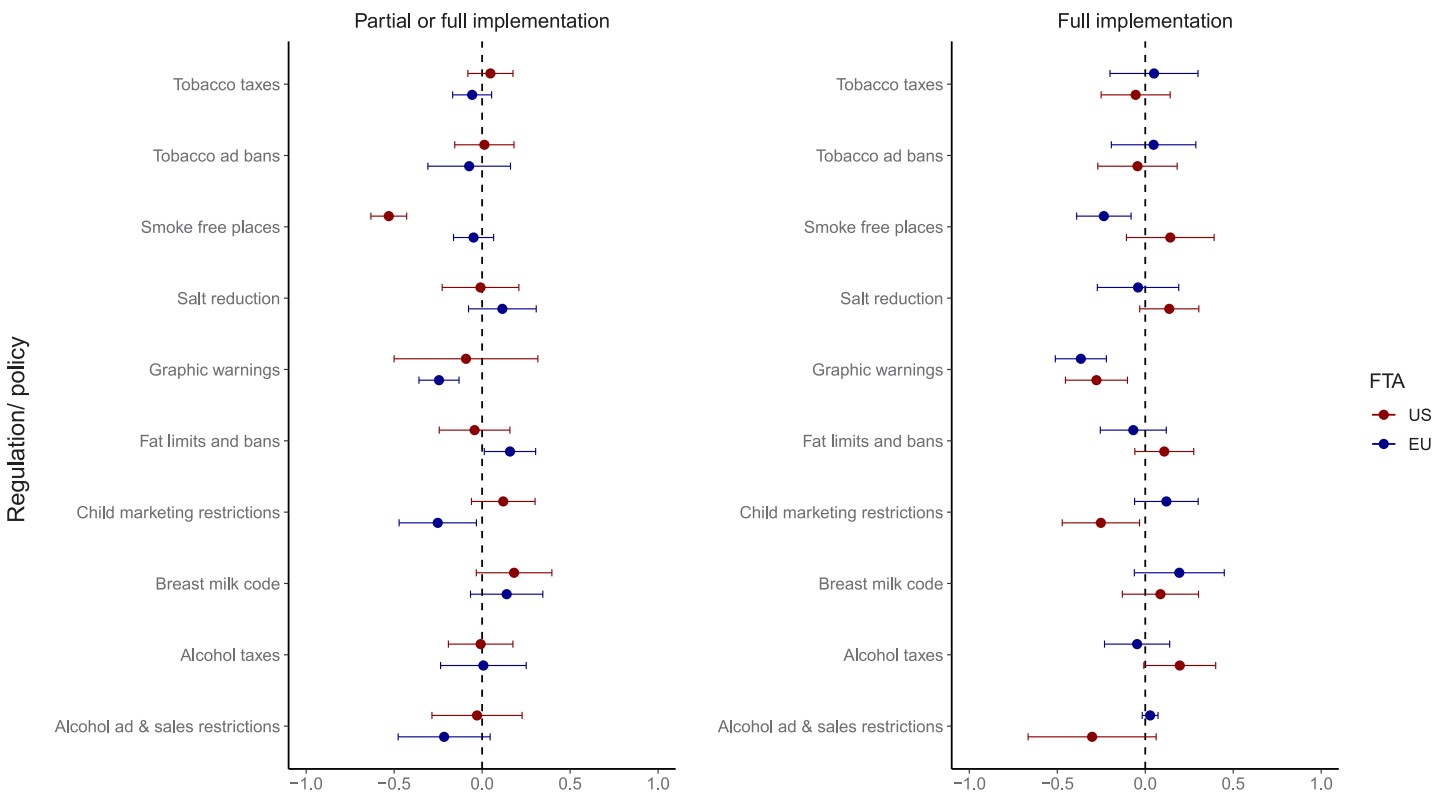

**Fig 4. AME of US and EU FTA participation on the predicted probability of partial or full implementation and full implementation of unhealthy commodity policies after matching.** Notes: Figure shows estimated average marginal effects with 95% CIs. AME, average marginal effect; CI, confidence interval; FTA, Free Trade Agreement.

Information shows that participating in US or EU FTAs was not significantly associated with reduction in the predicted probability of these policies. This bolsters confidence that our results are attributable to US/EU FTAs rather than alternative processes or country characteristics, which lead to reductions in NCD policy implementation in countries with US/EU FTAs, independent of any influence of FTAs.

We further examined whether our results may apply to all FTAs with countries where large producers of unhealthy commodities are headquartered, rather than US and EU FTAs specifically. Table G in S1 Supporting Information shows that countries with FTAs where large tobacco companies are headquartered are significantly less likely to achieve full implementation of tobacco advertising restrictions (AME = −0.3, 95% CI: −0.61 to −0.18). Table G in S1 Supporting Information does not identify a consistent pattern of increased or decreased implementation of food and alcohol policies in countries with FTAs where large food and alcohol companies are headquartered, with differing associations across policy outcomes.

We additionally examined whether our results may apply to Bilateral Investment Treaties (BITs) with the US and EU members, and whether our results were consistent when adjusting for participation in these BITs, as they contain some of the clauses included in FTAs that can be used to challenge policies, although their scope is heterogeneous and the pathways to influence may differ. Tables H-K in S1 Supporting Information present the full results of these sensitivity analyses. In summary, US and EU BITs were not associated with reduced

implementation of unhealthy commodity policies, and the results presented in Fig 3 were mostly consistent when adjusting for US and EU BIT. The relationships between EU FTAs and partial implementation of child food marketing restrictions and fat limits and bans were not significant in these models.

Tables L and M in S1 Supporting Information show the results from additional robustness checks assessing whether our results are explained by other processes. We first assess whether our results are consistent when we adjust for the implementation of NCD policies other than those targeting unhealthy commodities, which may again capture the influence of factors influencing NCD policy more generally. We further adjust for the total number of FTAs that countries participate in, as these other FTAs may also create opportunities for unhealthy commodity producers to challenge policies. In a final model, we control simultaneously for participation in BITs with the US or EU, for all FTAs with other countries, and for EU FTAs in models assessing US FTAs and for US FTAs in models assessing EU FTAs. All results are consistent in sign for all models; however, some models lose statistical significance at the 5% significance threshold when controls for all types of BITs and FTAs are incorporated simultaneously as might be expected given the very large number of controls. Again, the relationship between EU FTAs and increased partial implementation of fat limits and bans was not consistent in any additional models.

## Discussion

Our analysis has shown that US and EU FTA participation is associated with a substantial reduction in the predicted probability of implementing several WHO-recommended NCD policies that target unhealthy commodities. Approximately 19% (9/48) of tests showed a statistically significant decrease in the predicted probability of at least partial unhealthy commodity policy implementation, while 2% (1/48) showed an increase, and the latter was not robust in additional analyses. On average, US FTA participation and EU FTA participation were associated with a 5% decline in the predicted probability of at least partially implementing the unhealthy commodity policy in question, but these averages were not statistically significant. However, we identified substantial changes in the probability of implementation for some policies. Specifically, US FTA participation was associated with a 37% reduction in the predicted probability of fully implementing graphic tobacco warning policies and a 53% reduction in the probability of at least partially implementing smoke-free place policies. The probability of fully implementing of smoke-free place policies was also 24% lower in countries with US FTAs. EU FTA participation was similarly associated with a reduced probability of implementing select policies, including a 28% reduction in the probability of fully implementing graphic tobacco warning policies and a 25% reduction in the probability of fully implementing restrictions on child marketing of unhealthy food and drinks. These findings were consistent in a large number of additional analyses and robustness checks.

Our study provides new insight into the relationship between US/EU FTA participation and (non)implementation of WHO-backed policies that seek to restrict the marketing, sale, and consumption of unhealthy commodities. US/EU FTAs acknowledge that governments have a legitimate right to regulate to protect public health. Furthermore, high-profile trade and investment disputes were raised against Australia and Uruguay's tobacco packaging legislation, but the policies were ultimately deemed consistent with the treaties cited in each case [40,41]. While health protections and high-profile examples of public health policies being upheld in trade disputes might be expected to bolster governments' ability to regulate, our study suggests that this is not the case for US/EU FTA participants. Instead, our analysis corroborates previous concerns that US/EU FTAs incentivise and/or enable companies to pressure governments

to delay proposed policies or implement alternative measures that do not directly impact their products [12,42].

Notably, no formal disputes related to WHO-recommended NCD policies were initiated during the time period of our study. However, a number of countries rescinded policies relating to tobacco and child food marketing restrictions [29]. These phenomena, together with our results, suggest that tobacco, processed food, and soft drink producers may be using relatively informal channels to exert pressure via US/EU FTAs, for example, via direct lobbying, threatening trade disputes, or using opportunities for input established in regulatory cooperation clauses. This possibility is bolstered by findings from prior small-N studies that have identified instances when unhealthy commodity producers used, or appeared to use, trade rules to exert pressure on policy-makers to change policies affecting their products, including those which appeared to be influenced in our study: tobacco legislation and restrictions to the marketing of unhealthy foods to children [23,43]. It remains possible, however, that the existence of "behind the border" restrictions on policy within US/EU FTAs is sufficient to deter policy-makers from proposing regulations altogether, due to fears an industry trade challenges and disputes.

Our study has important limitations. Our results should not be interpreted as definitively causal, as our quasi-experimental comparisons have important underlying assumptions. One is that the associations we identify are unconfounded after matching and incorporating regression controls [32]. Data limitations also prevented us from estimating longitudinal models assessing within-country changes in implementation before and after joining US/EU FTAs. However, matching on observed covariates also matches on or controls for unobserved covariates insofar as they are correlated with the observed in our models [32]. Furthermore, our results were robust in sensitivity analyses assessing potential alternative explanations. There may nevertheless be additional unobserved sources of confounding, leading to the masking of true effects or finding spurious associations. Residual imbalances in political integration and democracy after matching may also help explain the results for EU FTAs. The small sample size also limited our ability to assess effect heterogeneity, for example, by country-income level or FTA design. Finally, our study was not designed to isolate the specific mechanism through which US/EU FTAs limit unhealthy commodity policies. There are several possible processes, including stakeholder input via regulatory cooperation processes established in US/EU FTAs and relatively informal dispute threats.

Further research is needed to evaluate whether our results apply to other FTAs, and whether domestic political prioritisation of unhealthy commodity policies may counteract any influence of US/EU FTAs and associated industry pressure. Our results also indicate a need to investigate sources of heterogeneity in the associations we identified. For example, we identified variation in the relationship of US/EU FTA participation and the implementation of similar policies (e.g., advertising or sales restrictions) across different commodities. This variation might be explained by a wide range of factors, such as differences in the degree of contentiousness of a particular policy where it targets different commodities, and the novelty of the policy and existing implementation levels prior to our study period. We also identified variation in the association between FTA participation and the implementation of different policies within the same commodity category. This may be explained, for example, by differences in the ability of industry actors to craft arguments that relate different policies to FTA rules, and differences in the visibility of economic benefits of the policies in addition to health benefits.

Our findings have important implications for policy-makers seeking to accelerate progress towards regulating unhealthy commodities and achieving global targets to reduce NCD mortality. Our results suggest that FTAs currently under negotiation may constrain efforts to achieve NCD-related global health targets in partner countries. For example, several US and

EU FTAs are now under negotiation, including a US–Kenya agreement; UK accession to the Comprehensive and Progressive Agreement for Trans-Pacific Partnership, which was heavily influenced by the US (UK-CPTPP); a potential future US–UK deal; and EU agreements with New Zealand, the Philippines, Indonesia, China, and Australia [44,45]. FTAs exhibit variation in their design, and these differences will need to be considered when appraising the potential impact of future FTAs. Governments negotiating new deals now have a potential opportunity to ensure new FTAs are drafted in ways that do not empower unhealthy commodity producers to challenge tobacco and junk food marketing policies through careful drafting at the negotiation stage. Policy-makers can, for example, ensure that new FTAs empower governments to protect populations from their harms. There are several ways to achieve this, for example, by excluding investor–state dispute settlement mechanisms from agreements to prevent disputes or dispute threats citing these clauses, limiting unhealthy commodity producers' access to proposals for polices that regulate their products in regulatory cooperation clauses, and limiting the scope and definition of key investor protections that industry might appeal to [19]. WHO has a potential role to play in providing technical support to countries as they negotiate trade agreements and in providing a forum for Member States to share their experiences and highlight commonly used corporate tactics. Indeed, Member States specifically called on WHO to perform this role at the 2019 European workshop on strengthening NCD implementation research capacity, citing industry opposition as a central barrier to NCD policy implementation [46].

Our findings also have implications for existing US/EU FTA participants. Countries with these FTAs appear to encounter difficulties in regulating tobacco and child food marketing. However, industry references to clauses in US/EU FTAs can be invalid and may constitute attempts to limit policies affecting their products by appealing to aspects of trade law that are poorly understood by policy-makers. Governments should be aware of this potential conflation of vested interests with the interpretation of FTA clauses. Governments may also be better able to implement unhealthy commodity policies despite opposition from industry where they have access to legal experts that can identify invalid trade-related claims at an early stage, and where they minimise industry involvement in policy-making processes via regulatory cooperation and lobbying. Finally, the risk of industry threats might be minimised by strategically designing unhealthy commodity policies in ways that accord with US/EU FTA rules while maximising efficacy. Whether countries are seeking to mitigate impacts of existing US/EU FTAs or negotiating new agreements, effective cross-government cooperation between legal, trade, and public health officials will be essential to accelerate progress to implement unhealthy commodity policies.

## Supporting information

**S1 Supporting Information. Including Appendices A-B and Tables A-M.**
(DOCX)

**S1 STROBE Checklist. STROBE Checklist for observational studies.**
(DOCX)

## Author Contributions

**Conceptualization:** Pepita Barlow, Luke N. Allen.

**Data curation:** Pepita Barlow.

**Formal analysis:** Pepita Barlow.

**Investigation:** Pepita Barlow.

**Methodology:** Pepita Barlow.

**Project administration:** Pepita Barlow.

**Validation:** Pepita Barlow, Luke N. Allen.

**Visualization:** Pepita Barlow.

**Writing – original draft:** Pepita Barlow.

**Writing – review & editing:** Pepita Barlow, Luke N. Allen.

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
