## [Editor Report · Decision Letter 0]

4 Aug 2022

Dear Dr Barlow, 

Thank you for submitting your manuscript entitled "US and EU Free Trade Agreements and implementation of policies to control tobacco, alcohol, and unhealthy food and drink: a quasi-experimental analysis" for consideration by PLOS Medicine.

Your manuscript has now been evaluated by the PLOS Medicine editorial staff and I am writing to let you know that we would like to send your submission out for external peer review.

Please re-submit your manuscript within two working days, i.e. by Aug 08 2022 11:59PM.

Kind regards,

Callam Davidson

Associate Editor

PLOS Medicine

---

## [Decision Letter · Decision Letter 1]

28 Sep 2022

Dear Dr. Barlow,

Thank you very much for submitting your manuscript "US and EU Free Trade Agreements and implementation of policies to control tobacco, alcohol, and unhealthy food and drink: a quasi-experimental analysis" (PMEDICINE-D-22-02643R1) for consideration at PLOS Medicine. 

Your paper was evaluated by an associate editor and discussed among all the editors here. It was also discussed with an academic editor with relevant expertise, and sent to independent reviewers, including a statistical reviewer. The reviews are appended at the bottom of this email and any accompanying reviewer attachments can be seen via the link below:

[LINK]

In light of these reviews, I am afraid that we will not be able to accept the manuscript for publication in the journal in its current form, but we would like to consider a revised version that addresses the reviewers' and editors' comments. Obviously we cannot make any decision about publication until we have seen the revised manuscript and your response, and we plan to seek re-review by one or more of the reviewers. 

We hope to receive your revised manuscript by Oct 19 2022 11:59PM. Please email us (plosmedicine@plos.org) if you have any questions or concerns.

We look forward to receiving your revised manuscript. 

Sincerely,

Callam Davidson, 

PLOS Medicine

plosmedicine.org

Comments from the Academic Editor:

My main comment was that out of 10 policies and 4 contexts (40 different tests), only 7 were significant in the direction that the paper focuses on (one significant in opposite direction, all others null). Whilst the authors note in the discussion that "a need to investigate why impacts may vary across regulatory domains and country-partners", I'd like to see a bit more evidence/theory-informed speculation here. Why might it be that FTAs impeded child food marketing restrictions, but not tobacco ad bans or alcohol ad and sales restrictions (ie why does the impact on essentially the same policies differ between commodity)? Similarly, why might FTAs impede smoke free places, but not tobacco ad bans/taxes (ie why do they impact on some but not all tobacco control efforts)? I suspect this is something to do with both how different commodities are 'valued' in society (ie tobacco universally considered 'bad', alcohol less so); and the co-effects of different policies (ie taxes are revenue raising, not JUST health promoting).

Please structure your abstract using the PLOS Medicine headings (Background, Methods and Findings, Conclusions).

Please avoid assertions of primacy (“We conducted the first…”) in your Abstract Background. Instead, the final sentence should clearly state the study question.

Assertions of primacy are also present in the Discussion and should be tempered accordingly (e.g., “to our knowledge”, or similar).

In the Abstract Methods and Findings, please include the study design and main outcome measures.

In the last sentence of the Abstract Methods and Findings section, please describe the main limitation(s) of the study's methodology.

Please remove the Funding section from your Abstract.

Please include continuous line numbering throughout your manuscript to facilitate further review. 

Please place citations within square brackets and preceding punctuation (e.g., [1]). This also applies to your appendices.

Please organise and cite your appendices as outlined here: https://journals.plos.org/plosmedicine/s/supporting-information

The footnotes in Appendix 4 appear to be labelled incorrectly.

Please ensure that the study is reported according to the STROBE guideline, and include the completed STROBE checklist as Supporting Information. Please add the following statement, or similar, to the Methods: "This study is reported as per the Strengthening the Reporting of Observational Studies in Epidemiology (STROBE) guideline (S1 Checklist)."

Table 1: Please define Free Trade Agreement (FTA) in the legend. 

Figure 4: Please define the 95% confidence interval bars in the legend. 

Reference 26 (your protocol) is a preprint: please add [preprint] to the reference to identify it as such. Further details can be found here: https://journals.plos.org/plosmedicine/s/submission-guidelines#loc-references

Comments from the reviewers:

Reviewer #1: Thanks for the opportunity to review your manuscript. My role is as a statistical reviewer, so my review concentrates on the study design, data, and analysis that are presented. I have put general questions first, followed by queries relevant to a specific section of the manuscript (with a page/paragraph reference).

This manuscript examines whether participation in a free-trade agreement is associated with the implementation of policies that target tobacco/alcohol/unhealthy food and drinks. Data is at the country level from three time points (2014, 2016, 2019), policy implementation data comes from WHO country surveys (based on WHO recommended policies) and information on free trade agreements is from collaborative database. A range of additional covariates is available for secondary analysis of heterogeneity of the TIA effect. The WHO specific recommended policies includes tobacco and alcohol taxes and restrictions, graphic warnings on tobacco packing, legislation on breast-milk substitutes and policies that target sodium, sat and trans fats, and food marketing to children. In the main analyses, the outcomes are assessed as unimplemented or fully implemented (and with alternative coding considered in sensitivity analyses). The 'exposure' variable for the main analysis is binary (part of TIA/not part of TIA). In the main analysis a matching process (based on Manholobis distance of the matching variables) to limit confounding is used. Where balance wasn't achieved, the covariate was included in the main logistic regression model. The protocol is very detailed (and well explained), there are some deviations from this (detailed in app 5), these are quite reasonable and reflect changes required once features of the final dataset were apparent (e.g. not enough variation to get sufficient sub-groups for some secondary analyses). Limitations of the study design are well articulated in the discussion. 

How is Mahalanobis distance calculated for differences in the region each country belongs to? I am more familiar with Mahalanobis distance calculated for continuous variables and I wasn't sure how it was done here.

With the Mahalanobis matching approach, is there a straightforward way to assess common support, i.e. that there is sufficient overlap between the FTA groups in the covariates or are the countries with FTAs fundamentally different from those without? 

My usual area of expertise is far away from free-trade agreements - so the following question is probably more for my own understanding than any issue with the manuscript. Are the FTAs similar enough to be that we can represent them as a universal, binary covariate? I ask because it seems that mainstream media reporting around FTAs always seems to emphasise differences between FTAs (for my country at least the reporting is always that 'we were ripped off compared to country XX') and I am curious if perhaps FTAs are broadly similar and I have a distorted impression of reality.

P6, Paragraph 2. What criteria were used to decide between purely regression adjustment and matching (once it was clear first order models weren't appropriate)? 

P7, Paragraph 3. This could be my reading of it but it seems like the part:

 "FTAit is one of two FTA indicators in country i with coefficient B1: i) US FTA participation, or ii) EU FTA participation, lagged by 1 year (t-1) to allow for a delayed effect; we estimate separate models for US FTAs and EU FTAs." 

is contradictory, if there is an indicator variable for EU or US participation woudn't that make the variable unvarying if separate models for US/EU are run? 

With this model each country will contribute multiple 'rows' of data regression model, i.e. there are repeated measurements (of different times) of the same countries. Is this correlation accounted for by the block bootstrap procedure? Also - is this the 'simple' version of block/cluster bootstrapping where an entire block/country is selected in the sampling, or is this the more complex procedure where there is second level of sampling of years within the selected blocks? 

Also to confirm, is country the 'block' for the block bootstrapping? 

P19. Table 1. I would consider removing the p-values here, the rest of the table provides everything necessary to understand the differences.

Should the title of the table include "EU FTAs"?

Under the EU FTAs there is a typo - "differesnce" should be difference. 

P18. What was the minimum number per category you considered to be appropriate allow for sufficient statistical power? 

Reviewer #2: This is an excellent and informative paper. Well done. Before publishing there are some exceedingly minor changes to make:

p. 5 'first quant study' is a bold claim that needs nuance. A google search shows there have been lots of (non-peer reviewed?) 'studies', impact assessments and so forth in this space. Also next section suggests a few of these studies. Suggest paring back this claim or adding nuance.

Typo p. 10 'All results are consistent in sign for all models'. 'Significance?'

P. 12 'window of opportunity' is vague. What is the window and why has this come about?

Reviewer #3: This paper uses natural experiment methods to assess the association of US or EU Free Trade Agreement participation with the implementation of various health policies. The paper thoroughly assembles what data is available for countries as the unit of analysis, and uses a matching procedure to try and approximate an unconfounded comparison. Unfortunately, not many matched country-pairs make it through to the final analytical sample - a necessary cost of trying to remove confounding, and increase internal validity. 

There are 48 'tests' of whether FTA participation effects health policies: 12 policies, by 4 combinations of EU/US FTA and Full or partial/Full classifications. Figure 4 shows these 48 'tests' as a forest plot, correctly portraying the mixed findings and lack of an overwhelming pattern. That said, there is a pattern - 9 of the 48 'tests' (19%) have 95% CI excluding the null AND a finding of worse health policy implementation if the country is a participant of a FTA, whereas only 1 'test' shows better health policy. Averaged across all 48 tests, the average percentage difference was -4.4%. 

The study is clearly worthy of publication. And it will generate substantial discussion and debate - rightly so. This is the first paper to quantitatively test this hypothesis in a comprehensive manner - testing a hypothesis long suspected.

I have only a few recommendations for the authors in revision.

1. It is not clear on reading the abstract that 48 tests were run, than many were null, etc. An indication of the PATTERN of 19% of tests showing a statistically significant decreases, compared to only one (2%) an increase, would help. Moreover, as a more thorough sense of the whole pattern, giving the average effect (-4.4%) with a 95% CI would be useful (which may require some careful math to allow for correlations of the partial / partial or full classification). 

2. Perhaps point out in the paper that pulling out the 9 'positive' tests for profiling is a bit risky. To many this falls foul of a statistical testing approach. Hence my suggestions above to give the average effect size. Or alternatively, the percentage of 'tests' with the effect size stronger than -20%, between --20% to -10%, etc - to convey a focus on magnitude of the associations.

3. The matching in Table 1 does not look as good as the text of the paper implies, or the absolute standardized means in Fig 3. For example, the absolute standardized mean differences in GDP in Fig 3 nicely moves almost to nil after matching - but not in Table 1. (Actually in Table 1, the differences in GDP remain, just flipped in sign.) Either I have misunderstood Table 1, or there is perhaps an error in the data in Table 1?

4. Box 2's title implies this is a WHO categorization - but it is actually the categorization used for analysis in this paper. Perhaps make this clearer. 

Congratulations to the authors on an important paper.

Reviewer #4: This paper shines a light on an important issue and methodologically speaking, is executed superbly. The authors write in an accessible and clear way about a highly complex and technical public health issue. My one, very minor, comment is that the authors might consider including the UK-CPTPP deal as an important US led agreement under negotation ( bottom of pg. 4 )

[LINK]

---

## [Decision Letter · Decision Letter 2]

4 Nov 2022

Dear Dr. Barlow,

Thank you very much for re-submitting your manuscript "US and EU Free Trade Agreements and implementation of policies to control tobacco, alcohol, and unhealthy food and drink: a quasi-experimental analysis" (PMEDICINE-D-22-02643R2) for review by PLOS Medicine.

I have discussed the paper with my colleagues and the academic editor and it was also seen again by two reviewers. I am pleased to say that provided the remaining editorial and production issues are dealt with we are planning to accept the paper for publication in the journal.

[LINK]

We look forward to receiving the revised manuscript by Nov 11 2022 11:59PM.   

Sincerely,

Callam Davidson, 

Associate Editor 

PLOS Medicine

plosmedicine.org

Requests from Editors:

The URL in your Data Availability Statement leads to an error page - please correct or confirm this link will be made active upon publication.

Please update your Author Summary to use 2-3 single sentence bullet points per question (please see here for further guidelines: https://journals.plos.org/plosmedicine/s/revising-your-manuscript).

Please check the formatting of your References (including Supplementary) is consistent (e.g., list first six authors before use of et al.) and follows our guidelines (https://journals.plos.org/plosmedicine/s/submission-guidelines#loc-references). 

Comments from Reviewers:

Reviewer #1: Thanks for the revised manuscript and responses to my review. The updates to the manuscript resolve all my original queries. A great piece of work.

Only one very small change - 'per-protocol' and 'protocol deviations' both have specific meaning to in clinical trials which are different to how they are used here. It might make S2 Appendix clearer to describe these as 'changes from pre-registered protocol' or 'according to pre-registered protocol'.

Reviewer #3: My comments have been satisfactorily addressed

[LINK]

---

## [Editor Report · Decision Letter 3]

22 Nov 2022

Dear Dr Barlow, 

On behalf of my colleagues and the Academic Editor, Professor Jean Adams, I am pleased to inform you that we have agreed to publish your manuscript "US and EU Free Trade Agreements and implementation of policies to control tobacco, alcohol, and unhealthy food and drink: a quasi-experimental analysis" (PMEDICINE-D-22-02643R3) in PLOS Medicine.

PRESS

Sincerely, 

Callam Davidson 

Associate Editor 

PLOS Medicine